# Decision-Making Mechanism of Joint Activities for the Elderly and Children in Integrated Welfare Facilities: A Discussion Based on “Motivation–Constraint” Interaction Model

**DOI:** 10.3390/ijerph191610424

**Published:** 2022-08-21

**Authors:** Wenjing Luo, Zhi Qiu, Yurika Yokoyama, Shengyuan Zheng

**Affiliations:** 1Institute of Architectural Design and Theoretical Research, Zhejiang University, Hangzhou 310058, China; 2Center for Balance Architecture, Zhejiang University, Hangzhou 310027, China; 3Graduate School of Arts and Sciences, The University of Tokyo, Tokyo 153-8902, Japan; 4The District Government of Binjiang, Hangzhou 310052, China

**Keywords:** integrated welfare facility, joint activities of the elderly and children, motivation and constraint, decision-making mechanism of joint activities

## Abstract

In China, joint activities for the elderly and children in integrated welfare facilities lack systematic decision procedures. By learning from the “leisure constraint” theory, the study puts forward six influencing indicators of motivation and constraint in the aspects of preliminary coordination, activity space and effect. By using semi-structured interviews and questionnaire surveys analyzed by deviation value computation, the study analyzes the evaluation value of influencing factors in the decision procedure of potential activity cases, where administrators and nurses act as two decision makers. Further, it discusses the decision-making mechanism based on the “motivation–constraint” interaction model. Firstly, it analyzes the dominant forces in the decision procedure, which are “motivation oriented”, “negotiation oriented” and “constraint oriented”. Secondly, it reveals that administrators and nurses as two decision makers tend to give positive motivation evaluations and deliberative constraints evaluations, respectively. Additionally, it analyzes the decision procedures of activities with distinct feasibility differentiation. Thirdly, it positions the levels of occurrence potential as “should occur”, “occurred but should be improved”, “potentially could occur” and “hard to occur”. Eventually, it analyzes the requirements and potential for joint activities under different service modules, which provides a theoretical foundation for the systematic planning and development of the joint activities.

## 1. Background

### 1.1. The Background and Problems of the Joint Activities of the Elderly and Children

Since the reform and opening up in 1978, the economy in China has grown rapidly along with a fast urbanization rate; as a result, the quality of public services has increased drastically [1]. Welfare organizations and facilities set up for vulnerable populations, including the elderly, orphans and disabled people, have also developed to new heights in recent years. In the economically developed cities, there is a trend of public welfare facilities, such as nursing homes and orphanages, being shifted from the city’s central region to the close suburbs constructed together due to the further centralization of the land and resources [2]. Furthermore, the child abandonment rate has decreased significantly, as overall living standards have improved. As a result, some orphanages have been incorporated into welfare facilities parks and renamed as Children’s Group, which has aided the development of the integrated welfare facility model. For example, there are 50 integrated welfare facilities that have been built in Zhejiang Province, and only Hangzhou, its provincial capital, has two large-scale integrated welfare facilities with at least 100 children beds (Figure 1).

In the aspect of space condition of the welfare facilities built in the integrated mode, although there is a “joint-construction” type where different facilities are built in one single park and another type called “adjoining” where different facilities built in separate parks closely adjoin, actually, by their nature, under the positive condition of joint management and adjacent location, the intimate living spaces of the elderly and children will stimulate the generation of joint activities between these two groups. From observation, these joint activities do not mean the elderly and children live together in one space on a long-term basis; however, most of the time, these two age groups of people are taken care of in their own functional spaces and carry out basic daily activities. Moreover, planned by the facility staff at a certain frequency, these joint activities are held randomly by borrowing some spaces in the elderly or children’s own functional spaces. Normally, the orphans attend activities under the facility staff’s guidance, and the elderly attend activities on their own. Additionally, since most of the joint activities have strong communicative attributes, most of the elderly who are willing to attend the joint activities are in a good physical and mental condition, as well as with a high sense of social identity and family cognition. Compared with the basic elderly security provided for high-risk families (“Wubao” families) by the government before, in recent years, the private capital has been gradually involved in the operation of these welfare facilities. Further, the relative scope of services of these facilities has expanded, whereby they also accept the elderly from normal families under the support of the government, improving the average mental and physical health quality of the elderly [3]. Additionally, the participation rate and experience of these joint activities have been improved, which also benefits the sustainability of the operation of these activities.

At the time when the nursing homes and orphanages are constructed separately, these facilities have clear, strict design guidelines and construction standards with supporting policies [4,5]. However, these standards have not been updated so far, and they lack the corresponding contents for the joint activities of the elderly and children. This is blamed on the absence of a widely accepted elderly–children activity system as a reference at present, so the activities occur randomly in an unstable way. As a result, although both nursing homes and orphanages have already standardized their construction systems in detail [6,7], there is still no clear public space construction system or design approaches for spaces for the joint activities of the elderly and children. These randomly held elderly–children’s joint activities mostly make use of the surrounding outdoor space in the welfare parks or a temporary requisition of multi-functional rooms in the original nursing homes or orphanages. However, such “temporary” spaces are difficult for supporting specific activities and always lead to a negative effect on the participants’ experience and safety.

### 1.2. Research Significance of the Joint Activities of the Elderly and Children

There is a dynamic balance system between the integrated welfare facilities having basic services and adequate space and the people living there. The primary purpose of promoting the joint activities in various functional modules of the facility is to encourage interaction between the two groups in order to foster a positive relationship with social attributes [8]. The core design logic of the space as the activity carrier derives from the “elderly–children joint activity” itself (Figure 2). Therefore, it is critical to clarify the appropriate system. The quality and characteristics of elderly–children operation systems and activities vary in different countries around the world. Even though these cases could be used as a guideline for the design of integrated welfare facilities in China, their applicability and adaptability in the new environment must be scientifically proven. Therefore, the decision-making mechanism behind them should be clarified to analyze the appropriate types of joint activities. 

This research distills the experience of the joint activities of the elderly and children in different countries. Additionally, based on the occurrence mechanism, the paper reveals a decision-making mechanism in the integrated welfare facilities in China, screening the appropriate activity types for integrated welfare facilities. The paper also explains the requirements for holding various activities and the corresponding methods for tackling difficulties. The findings provide a theoretical basis for further program planning and architecture design of the integrated welfare facilities. Further, this decision-making path under the positive and negative forces offers a reference for the following operation of facilities.

## 2. Literature Review and Methods

### 2.1. The Introduction of the Leisure Constraint Theory

People make various types of decisions [9]. The decision-making system contains three important elements, which are the decision makers, the influencing factors that affect the decision-making process and the results of the decision [10]. Due to the difference between the decision makers and the influencing factors, the results of the decision have a big difference. The most common type in various decision-making systems is the one with only one decision-maker group, which creates the influencing factors—the key elements that affect the result. To better optimize, screen and eliminate the decision options, scholars have different strategies, including screening the dominant options from the vast decision options [11], comprehensively evaluating the positive and negative influencing factors [12] or picking the typical drawbacks and options with problems [13]. 

Most decision-making systems contain various types of decision makers. Additionally, one situation is that these decision makers have a contradictory sense of values, which leads to different influencing factors, such as decision-making processes in large management teams, for instance [14]. To figure out this problem, some scholars come up with theories and methods, called the multi-perspective strategic decision making. In other situations, there are different decision makers in the system, and these bodies have a consistent sense of values and goals, and here are two situations in the aspect of influencing factors. One is that influencing factors have positive effects on the result of the decision. For instance, in the research of stimulating the human–computer interaction by inner and external initiatives, some scholars bring forward a theory and method called the motivational reward framework for affective agents [15]. The other is that there are interaction forces between motivation and constraints; for example, when tourists make decisions on whether they should travel to a place, scholars bring forward the leisure constraint theory to discuss the positive and negative influencing factors of participating in the activities. Additionally, this theory analyzes the extent of the influencing factors affecting the results of different factors [16,17,18].

The joint activities for the elderly and children will benefit the physical and mental health of the elderly and children and raise the quality of the services for these two age groups, which also form a virtuous social order in the welfare facilities. Therefore, generally, the organization of the joint activities of the elderly and children is supported by various groups of people, including the operator of the facilities and the participants. However, the decision-making process of the joint activities is motivated by the willingness of the elderly and children’s groups and the positive social effects these activities make, and they are constrained by the limited funding and safety concerns. So, there is a vague decision-making process under these two positive and negative forces. As a result, it is necessary to examine both the positive and negative aspects to estimate the feasibility of the joint activity. 

The joint activities of the elderly and children in integrated welfare facilities have three core attributes, which are “multiple decision makers, consistent goals, positive and negative influencing factors”, which is consistent with the leisure constraint theory. The leisure constraint theory covers various types and a wide range of discussions. Most typically, the one in tourism studies is aimed to figure out the influencing factors of temporary activities during a given trip organized by the tourists or group relaxation [19,20]. Moreover, some scholars investigate long-term factors that affect the participation in activities aimed at high school students in different grades [21], adults in different life cycles [22] and one group of people in a different time period [23]. Since the joint activities in integrated welfare facilities exist on a long-term basis, the decision-making mode of this research is closer to the latter research mode, so this research brings in the typical model of the leisure constraint theory to discuss the decision-making process of the joint activities of the elderly and children in integrated welfare facilities.

In this model, “motivation” is the positive influencing factor that stimulates the occurrence of the joint activities [24], including demands, reasons and satisfaction of the activities [25], which are generally regarded as homogenized positive incentives [26]. On the other hand, “constraints” are factors restricting the occurrence of the activities, which are divided into limitors and prohibitors [22] depending on the extent of the influencing factors affecting the activities. Restrictive constraints will only restrict the occurrence of the activities to some extent instead of fully canceling the activities; however, when prohibitive constraints reach some degree, the activities would not occur at all [27]. However, the feasibility of activities is not bound to decrease when facing constraints because appropriate negotiation would counteract the negative effect that constraints create. According to the model of the leisure constraint theory [28], Jackson emphasizes the impact of negotiation instead of the result of participating or not [29]. In the research of leisure constraint negotiation, some scholars use regression and structural equation modeling programs to test four models, which are independence, negotiation buffer, constraint effects mitigation and perceived constraint reduction [30], which proves motivations’ and constraints’ influence on the leisure negotiation and feasibility of activities.

### 2.2. Sample Extraction of the Joint Activities of the Elderly and Children

Throughout the world, especially in economically developed countries, various types of cases of joint activities of the elderly and children have emerged under the pressure of an aging society, including some practices in China. To discuss the adaptability degree of each case in China’s integrated welfare facilities, through various types of methods, including the literature review, field observation and expert interviews, this research collects relatively joint activities cases and builds a potential database of joint activities of the elderly and children. This research learns from the “motivation–constraint” decision-making model in the leisure constraint theory to estimate and screen the joint activities by which it provides a feasible case support for the joint activities of the elderly and children in integrated welfare facilities.

Instead of being managed in a top-down mode controlled by the Chinese government, the joint activities of the elderly and children mainly come in two types—namely, institutionally joined or joined in communities. Institutionally joint activities widely appeared in the United State and Japan. Apart from combining facilities as a whole for both age groups or constructing them adjacently, some will combine the normal public space, such as activity playgrounds and libraries, together [31]. Moreover, some non-governmental organizations build “Toyama type” facilities, which are constructed in a multi-functional, integrated and small-scaled mode. Further, they used the concept of “symbiosis beyond generation” [32] and the principle of “effective social resources utilization” [33], and then, they developed daily joint activities [34,35,36], such as some cultural educational events, including local countryside culture inheritance, parenting lectures, concerts, flower tours and so on [37,38]. Apart from that, they developed a series of outdoor activity projects, such as picnics, kiting, collecting, hot spring, pilgrimage and so on [33,39]. The high-end nursing institutions for the aging in the United State have a mode called the inter-generational learning center, which offers the elderly and children learning activities, including skills acquirement, and a sense of value and knowledge [40]. In Germany, joint activities of the elderly and children in the community are led by the government or non-governmental organizations. Through children’s education, parenting lessons, family support and youth counseling, they provide various types of space and services support [41].

The experience of joint activities in other countries produced by the joining of the elderly and children can be references for this research, but still, it is not applicable to China’s integrated welfare facilities. This paper distills the activities recorded in the related literature and adds the cases observed in the integrated welfare facilities as a complement. Additionally, this paper screens the activities according to the cultural and social context of China as the first step and categorizes them into five service-supporting modules, which are daily support, medical care, education and culture, social practice and entertainment [3], and their corresponding functions and spaces. Then, a basic information database of the joint activities of the elderly and children in integrated welfare facilities is formed (Table 1).

### 2.3. The Decision-Making Framework Development of the Joint Activities for the Elderly and Children in Integrated Welfare Facilities

There are different groups of people involved in integrated welfare facilities, and it is necessary to look at the operators of the facilities and the elderly, as well as the children, when choosing who the main body of decision makers are. In the traditional leisure constraint theory, the participants of the activities are the body of decision makers with independent willingness [42], and an evaluation system cannot be formed by estimating only one activity’s influencing factors. However, in the decision-making framework this research constructed, the orphans are supervised by the childcare workers, who should not be regarded as part of the decision-maker groups, without the ability to decide whether to take part in the activities or not [43]. On the other hand, the elderly have an intensely subjective preference to choose one single activity, so only their preference will be considered as one of the influencing factors. In the context of China, the planning, construction and operation stages of the welfare facilities are all strictly controlled by the government, which reflects the top-down operation mode that follows “Orders from the government-Executed by the facility-Feedback from the facility-Orders from the government”. Therefore, the management administrators in the welfare facilities who represent the willing from the government side plus the employees tackling the special affairs are the key parts of the planning and execution of the joint activities of the elderly and children. Moreover, some of the administrators in the facilities are appointed by the government who could also represent the direct perspective and willingness from the government side. Their duty is the daily operation of the management and administration, and they will regularly report to the relative department in the government, which means they have a relatively strong power of influencing the decision and the final results in the aspects of policy making, budget and human resources management. Therefore, the administrators are part of the decision-making body who represent the perspective and the willingness of the government. Additionally, nurses are the decision-making body who have close relationships with both age groups, and they are also the people who have precise information about both age groups’ health condition, so nurses are the decision-making body who represent the perspectives of the demands from the two age groups and the real condition of the activities.

Based on the analysis above, this paper brings forward a basic structure decision-making framework consisting of activity cases, decision makers, motivations and constraints (Figure 3). The research investigates the level of motivation and constraints according to the evaluation of different sub-items and explores the relationship between the different influencing factors and feasibility. Based on this, the research compares and studies the evaluation difference between the administrators and nurses. There are several factors that affect the motivation and constraints of organizing the activities, but if the decision-making system is too complicated, then it will slow down the action of information collection, explanation and conversion, which will constrain the decision making based on the influencing factors [44]. However, this research is constructed based on a survey of experts, which focuses on choosing the cases of activities, so that it is better to control the number of motivation and constraint factors and choose them accurately.

The activities organized in a welfare facility are divided into three levels—the planning phase to the implementation phase—which are the preliminary planning level, considering a comprehensive perspective, the space level, involved in the processes of pushing the planning phase forward to the implementation phase [45], and the effect level, which is presented in the final implementation stage (Figure 4). At each level, there are both motivation and constraint factors that act as the main influencing factors. 

In the aspect of motivation, firstly, in the early planning phase, the interaction between the elderly and children should be widely encouraged by the society, so that it can be promoted during the implementation, and the corresponding policies are developed to support its development. Secondly, in the process of preparing the activities, the spaces are contained in the spaces within one particular park, which provides accessible space conditions for the joint activities of the two age groups. Thirdly, at the effect level, to improve the life quality in the welfare park, both the psychological and physiological aspects of the elderly and children require more positive stimulation from various types of activities. In addition, other factors, such as the promotion effects for the facilities, will also have a positive impact on the joint activities.

In the aspect of constraints, firstly, in the early planning phase, it is necessary to deploy more workers for activity management compared to during normal time. However, due to the shortage of nursing labor in China, inevitably, the administrators and volunteer teams from outside facilities should temporarily assist in the activities’ operation. Secondly, there are potential risks during the preparation phase of activities if the spaces lack professional equipment and barrier-free designs [46]. Thirdly, the financial support provided by the government for welfare facilities is always limited, so the types and frequency of activities that could be implemented are constrained by the budget [47]. Additionally, it is worth mentioning that the number of people attending the activity as well as their physical health and other criteria will also affect the interaction effect of activities. This research summarizes the main factors of motivation and constraints that promote or prevent the joint activities in Table 2.

This study brings forward three hypotheses about the decision-making mechanism of the joint activities based on the research above.

**Hypothesis 1** **(H1).**
*There is differentiation in the decision-making mechanisms of the elderly–children activities in the “Daily Support, Medical Care, Education and Culture, Social Practice and Entertainment” service modules in the integrated welfare facilities.*


**Hypothesis 2** **(H2).**
*The degree of each motivation and constraint factor influencing the joint activities of the elderly and children in integrated welfare facilities varies from administrators to nurses.*


**Hypothesis 3** **(H3).**
*The evaluation standards and the validity of evaluation vary from administrators to nurses.*


### 2.4. Data Acquisition and Analysis Methods

To avoid the differentiation of the decision-making bodies and other decision-making factors of the elderly and children activities caused by the different regions of welfare facilities, this research chooses two typical integrated welfare facilities in Hangzhou (Figure 5), which are the Xiao Shan Welfare Facility (XSF) and the Yu Hang Welfare Facility (YHF). Although in the aspect of space condition these two integrated welfare facilities, respectively, belong to the “joint-construction” type and the “adjoining” type and vary in their sizes, they both adhere strictly to the integrated welfare facility architecture design standard at the administrative district level. Additionally, after construction, both of them are under the jurisdiction of the same government sector, which is the Civil Administration Office in Zhejiang Province. Additionally, there is a certain degree of similarity between the groups of people, including the elderly, children, administrators and nurses. Furthermore, both of them have a certain number of joint activities that occur with a certain frequency (Table 3).

By using the Likert rating scale questionnaire, the research scores the six influencing factors under the decision-making framework of joint activities, ranging from 1 to 4, which represents strongly disagree to strongly agree, respectively. The survey was carried out by using semi-structured interviews. Although there were a certain number of facilities’ staff, only a few administrators and nurses could participate in the decision-making process. Based on the comparison and analysis of two facilities, which reduced the decision-making difference caused by the facilities’ distinctions, this research added together the sample numbers from both facilities to keep a certain number of samples. In total, 55 copies of surveys were gathered. By qualitative examination of the data, 17 invalid questionnaire surveys were eliminated, which contained mistakes, such as mechanically repeated numbers or obvious inconsistency. Therefore, 38 valid surveys were collected (Table 4). (1) Different levels of facility administrators (*n* = 16) in different departments, such as management, finance and property management; (2) nurses (*n* = 22) who primarily worked in the nursing department, children’s department, infirmary and other medically related departments. The proportion between the administrators and nurses who participated in the survey was nearly in line with the proportion between them in the whole facility.

This questionnaire included two types of decision makers. Because of the differentiation in their duties and professional backgrounds, in the aspect of the joint activities of the elderly and children, the evaluation they offered had diverse opinions from a variety of perspectives, which corresponded to their authority and weight of influence [48]. In terms of estimating the evaluators’ authority, several rounds of anonymous surveys were used to examine the degree of the concentration, dispersion and coordination of expert opinion data based on the widely used Delphi method. The evaluation accuracy was based on their authority, while the entropy method, on the other hand, primarily determined the utility value of various expert group information based on the order difference of information contained in each index. However, the Delphi method is often used for formal situations and targeted issues, since it involves numerous rounds of investigation. The entropy approach only considered the information entropy without the examination of the work content and the responsibility of administrators and nurses. As a result, the evaluation authority of the two evaluation groups in different activities cannot be reflected by these two methods.

The data collected in this research analyzed the evaluation made by the two decision-making groups and compared each “motivation–constraint” element. Because the same numeric value in different series had a different corresponding level of degree, it is not accurate to evaluate the value in its series if we only compare the numeric value. Therefore, to evaluate and compare the score levels of different factors, this study used the deviation value method, which is one of the standardized methods. Based on the standardization of the Z-score, this method combines the mean and the standard deviation of the original data clusters into a single data set with a mean of 50 and a standard deviation of 10. Then, it becomes the comparison basis. Because this method can compare the relative level of each value in its group intuitively, it is widely used in examination evaluation in the education field. Under this standardized method, a value of more than 50 indicates the degree of this factor is higher than the average level. Factors with a deviation value above 60 are regarded as strong influencing factors of motivations or constraints.

## 3. Results

According to the calculation results of the deviation value, this study estimated the feasibility of activity in different categories and analyzed the tendencies of evaluation making by the administrators and nurses, as well as the activity cases that the two groups of decision makers thought were different.

### 3.1. Relationship between Feasibility, Motivation and Constraint Value

After standardizing the deviation value of the average value of motivation, constraint and feasibility, various activities were ranked from high to low in terms of their feasibility, as shown in Figure 6. It shows that all three factors of motivation values of each activity were relatively approximate, while there was sometimes a large gap between the different constraint values (for example, the degree of the capital budget was sometimes significantly higher than the other two types of constraint values), which indicates that decision makers tended to consider the motivation values of elderly–children activities as a whole as a prerequisite. However, they would come up with more target-specific analyses when dealing with the constraints encountered later.

Moreover, while most activities’ feasibility values were close to the motivation values, there were some cases in which the feasibility values were much higher or lower than the motivation values. As a result, there are three types of the activities’ “motivation–constraint” mechanisms that should explain the different categories.

#### 3.1.1. Motivation Oriented: Feasibility Value Is Close to Motivation Value

There are 19 types of activities whose motivation value is close to the feasibility value (Table 5). Regardless of whether the constraint value is high or low, the feasibility of these activities remains remarkably consistent with the level of motivation values. Those activities focus on high-frequency, daily and regular entertainment and health care, indicating that the initial motivation in the planning stage is sufficient for these activities, and if motivation is insufficient, the activity is not necessary to be carried out, which is the most conventional and straightforward activity decision-making mode.

#### 3.1.2. Negotiation Oriented: Feasibility Value Is Significantly Higher than Motivation Value

There is a specific type of activity whose feasibility value exceeds the level of motivation (Table 6). Additionally, there are two cases leading to this type of situation. Firstly, despite the fact that such activities are hard to implement and require more supporting facilities, they have a positive effect on the welfare facilities’ image, which benefits the long-term development. For example, these types of classes include educational introspection lessons, gardening, singing and dancing activities, exhibitions and so on. Secondly, the elderly and children have outstanding motivations toward some particular types of activities. For example, both age groups present a particularly strong demand for the Skill Show held during the Spring Festival, which makes the planners overcome the constraints.

#### 3.1.3. Constraint Oriented: Feasibility Value Is Significantly Lower than Motivation Value

The number of activities dominated by constraints is the lowest within the joint activities, containing only six samples (Table 7). These activities are all in the field of professional medical treatment. The constraints in these types are difficult to overcome by welfare facilities due to the limitation on the resources and the profession. As a result, while the results of motivation evaluation indicate that such activities are critical in welfare facilities, most of them are difficult to conduct due to the prohibitive constraints.

### 3.2. Evaluation Tendency of Administrators and Nurses

There is a significant difference between the administrators’ and nurses’ original values of motivation and constraints (Figure 7). The values of nurses’ evaluation on motivation and feasibility mostly gathered in a range from 3 to 4, and the overall values of the average feasibility values were higher than the administrators’ evaluation values, which shows that nurses have a more optimistic attitude toward those activities. Additionally, the nurses’ evaluation of constraint values was more dispersed than the distribution of the ones of the administrators. As the two types of decision makers have different evaluation ranges for the activities, it is difficult to compare the values directly. Therefore, it is necessary to conduct a deviation value computation and compare the values of activity motivation, constraint and feasibility based on their evaluation levels.

Three-dimensional scatter is a form of stereogram that illustrates the statistical relationship between pairs of variables. The X-axis represents the deviation value of feasibility, while the Y-axis represents the level of deviation for six different types of motivations and constraints. In Figure 8, most of the data points of the administrators’ and nurses’ motivation and constraint values are concentrated areas, with only a small portion of the data “deviating” from most of the data set. The tendency of the evaluation of motivation and constraint values made by the administrators and nurses also has a similar increase along with the rise in feasibility. Therefore, the two groups have similar evaluations on the interaction relationship between the “motivation–constraint” factors and feasibility. However, the consistency of the nurses’ evaluations of the constraints and feasibility is more concentrated than the one of the administrators’, particularly in terms of the cases of low-feasibility activities. Meanwhile, the degree of association of the evaluation of activity constraints and feasibility made by the nurses is higher than the one of the administrators. This reflects the difference in the division of responsibilities of these two groups. Administrators can reach a more consistent view on the motivation evaluation in the decision-making process relatively easily, whereas nurses have a better understanding of the constraints and difficulties encountered during the implementation phase.

### 3.3. Differently Evaluated Activities by Administrators and Nurses

In terms of the evaluation ability of the joint activities, since they maintain consistency in the feasibility of most joint activities represented by the deviation values of those two being both higher or lower than 50 points, it means they hold the same positive or negative attitude toward the joint activity. However, there are five activities that show different evaluation results of the administrators and nurses (Table 8). One of them shows a positive attitude from the administrators versus an opposite one from the nurses; the others show a positive attitude from the nurses versus an opposite one from the administrators.

Firstly, regarding the skill show, the administrators think that there is good incentive for holding the event, while the nurses think the motivation value is not high. Despite the fact that the administrators recognize the difficulties in holding the skill show, especially in the aspect of the management difficulty, they believe these constraints can be figured out. Moreover, the nurses generally hold an opinion of high feasibility on the other four activities, which are the culture and skill course, workshop, physical examination and rehabilitation. Both the administrators and nurses agree that the cultural and educational activities, such as the first two, are simple to implement, but the administrators give a negative assessment on the motivation. While all decision makers recognize the high value and real demand for the latter two medical care activities, the administrators place a high value on their constraints, which results in the final negative feasibility assessment.

The result reflects the differences in the sense of the values of the administrators and the nurses to some extent. That is, administrators are more concerned about the effect of publicity, whereas nurses believe it is more important to satisfy the elderly and children’s daily needs.

### 3.4. The Classification and Characteristics of the Joint Activities

Based on the assessments made by the administrators and nurses, as well as the occurrence status of activities in the integrated welfare facilities, the activities in the database are categorized into four types according to their difficulties in implementation (Table 9). Furthermore, to evaluate their assessment accuracy, the researchers compare the overall assessments made by the administrators and nurses with the current situation of the implementation of joint activities.

#### 3.4.1. Type A—Activities “Should Occur”

These activities are already implemented in practice and recognized by both administrators and nurses; they can be organized on a long-term basis in a short time duration with a high frequency (Table 10). These activities are mainly daily and periodic events without excessive capital budget or difficulties. Most of these activities take place indoors. Moreover, a few activities are held outdoors, but they require no complex equipment and can be held for the elderly and children frequently.

#### 3.4.2. Type B—Activities “Occurred but Should Be Improved”

These activities were already implemented, but either or both two groups of decision makers gave them a low feasibility score (Table 11). They primarily consist of large-scale recreational or daily life activities and have a relatively high requirement and high demand to be carried out. However, due to the constraints in the aspects of finance, safety and management, the quality of these activities is low, which results in low feasibility values. It is vital to assess the feasibility of such operations and carry out proper preparations accordingly.

#### 3.4.3. Type C—Activities “Potentially Could Occur”

Such activities have not yet been carried out, but either or both two groups of decision makers gave them a high feasibility score (Table 12). That is, while they have recognized the value of these activities, they are still not included in the previous plan, so they can be planned for in the following activities in the planning phase. However, since they have not been held before, it may be necessary to prepare new sites, equipment and other required materials and facilities.

#### 3.4.4. Type D—Activities “Hard to Occur” 

Activities that have not yet taken place are rated negatively by the administrators and nurses (Table 13). There are various reasons behind this, including high security risk, high requirement for professionals or low necessity. Although some of the activities in the database have been held in other kinds of elderly–children institutions, they are not appropriate for the current welfare facility environment; therefore, these activities are not discussed in this paper.

## 4. Discussion: Feasibility of Activities in Different Service Modules

Based on the results above, all activities are re-organized and classified according to the corresponding service modules to discuss the development potential and types as well as the authority level of different decision makers (Table 14).

### 4.1. Daily Support Module

In this sector, both groups of decision makers maintain constant evaluation. This module contains mainly long-term daily activities. Most of them are classified in “Should Occur”, “Occurred but Should be Improved” or “Potentially Could Occur” categories, which are more suitable for activity planning and implementation, and only Consulting was regarded as unpractical among them. As a result, more low-cost, high-frequency and low-communication activities for the elderly and children can be planned for in the future.

### 4.2. Medical Care Module

In the medical care module, the evaluations from the nurses played a dominant part, and they were more accurate. The medical care service activities are highly professional, so if they fail to meet the requirements, the activities could not be held due to the space limitation. Additionally, the activities would be exposed to a number of potential security issues even in a qualified space. Further, the expense of daily maintenance of the health care equipment would lay a heavy burden on the budget of integrated welfare facilities. As a result, half of these activities are currently difficult to implement, and only daily exercise can be carried out directly. Since children and the elderly have a rigid demand for medical care, this module should be considered as an alternative plan, and it is necessary to make an effort to set up policy support and improve this type of activity in terms of space, management, funding and so on.

### 4.3. Education and Culture Module

These activities are mostly cultural courses and handicrafts, but a few of them were not held. This is because, firstly, educational activities for the elderly and children without family members are not an essential. Secondly, some specific activities, such as exhibitions and cultural courses, have high requirements for the space condition and, in particular, support workers. Additionally, the activity “Reading on the internet” receives a high level of feasibility evaluation because it can achieve both educational and entertainment goals. However, there is no particular space built for this kind of activity yet, so it is worth to be taken into consideration in the future. 

### 4.4. Social Practice Module

This part mostly depends on the decision opinions made by the administrators. Except for some high safety risk activities, such as childcare, other activities gained strong support from either or both two groups of decision makers. Because they are not complicated to implement and have a good publicity effect, the administrators are more likely to promote and organize these activities. Moreover, welfare facilities are advised to organize activities that will satisfy both the social practice and the entertainment need in the future as a way to extend the range of activity types.

### 4.5. Entertainment Module

The entertainment module contains a wide range of activities, and most of them have already been carried out. The others are too difficult to hold. That is to say, as long as there are no unconquerable prohibitive constraints, entertainment activities will be carried out. Among them, two types of activities in the “Occurred but Should be Improved” category are outdoor activities. At present, the outdoor recreational spaces in integrated welfare facilities still have visible flaws, which become a constraint for outdoor activities.

## 5. Conclusions

There are differentiations of social circumstances, social attributes and welfare policies between each country; therefore, the joint activities of the elderly and children could be a reference for China instead of a guideline. Therefore, it is necessary to scientifically screen and analyze precisely the decision-making level of the activities, so that these existing cases can be applied to integrated welfare facilities and guide the subsequent architectural design. Because the joint activities are generally supported by the operation groups in welfare facilities, and the decision-making process is influenced by the positive and negative factors, this paper examined the leisure constraints models in tourism. It quantified and discussed the motivations and constraints behind the decision-making process of the joint activities of the elderly and children. It collected the joint activities cases and built a potential database of them, screened according to the decision-making model. Additionally, in this decision-making model of the joint activities of the elderly and children, both the management administrators representing the perspective of the government and the supervisors together with the nurses with perspectives from the users and real conditions act as influential components of the decision making process, and they are regarded as the main body of the decision makers. The decision makers should not only take the real mental and physical demands of the elderly and children into consideration but also think about and control the resources of spaces, human labor and capital budget. Therefore, the research distilled three motivation factors, namely, public attitude, spatial support and strong demand, as well as three constraint factors, namely, management difficulty, potential risk and capital budget, and also evaluated the feasibility of each activity. Based on this, the paper executed “motivation–constraint” models of the decision-making process. When comparing the motivation and feasibility values, it found that the motivation values were mostly consistent with the feasibility values. However, the evaluation of the motivation value and the feasibility value of large-scale entertainment and professional medical care activities had a big difference between them. So, three different motivation–constraint influencing models were summarized based on the analysis above. Moreover, the nurses tended to give higher scores on motivation and feasibility compared to the administrators, so their numerical evaluations presented a big difference. To compare the values of the same standard, the paper produced standardized deviation values and then reached a conclusion that the administrators and nurses gave similar evaluations regarding most activities, except for several ones based on a different sense of value. Based on the evaluation made by both groups and the occurrence conditions, this paper classified the activities into four categories, which were activities that “Should Occur”, “Occurred but Should be Improved”, “Potentially Could Occur” or were “Hard to Occur”, and each type had the corresponding requirement for planning. Finally, all the activities were categorized into five service modules to evaluate the potential for development and the limitation of each type of activity in these five modules. Therefore, this provided a theoretical foundation for program planning and space construction from a top-down decision-making perspective.

The mode of construction and operation of integrated welfare facilities is a symbol of the level of welfare development in China. Even though welfare programs in various countries are constantly improving, vulnerable people, including the elderly and children, can never be ignored. Therefore, the integrated welfare facility model of China will continue to exist and be improved in the future. How to plan and implement joint activities of the elderly and children in welfare facilities that make them both happy and strengthen the personality of disabled children, as well as keep the sustainability of activities, will be a constant subject in the process of development of the welfare system in China. The planning mechanism of joint activities that this research aimed at is not only beneficial for constructing a complete activity system of the facilities, but it also provides a theoretical foundation for subsequent research of the joint space construction for the elderly and children. Especially in a situation of international pneumonia outbreak caused by the 2020 novel coronavirus disease (COVID-19), to avoid the spread of the infection, China’s integrated welfare facilities all adopted close-end management measures. This means that the social voluntary workers and even relatives are allowed to visit at a more limited frequency to different degrees. Additionally, this situation represents an even greater value of the joint activities in these facilities, which have a promoting effect on the elderly and children’s social relationships and social interaction. Therefore, this study will also provide new vision for the operators of welfare institutions or facilities in other countries, where most of the elderly and children joint activities take place under decision procedures.

However, although the welfare development progress in China has been acknowledged throughout the world, as a developing country, there is still a need for further model clarification, mechanism development, space construction and evaluation of management and operation. Therefore, mature conditions and opportunities for discussion on the elderly and children’s joint activities in welfare facilities are still lacking. Additionally, this is the main reason for this research to be recognized as “pioneering research”, which shows research limitations in several aspects. For example, although there is a certain number of staff working in integrated welfare facilities, only a few of them could participate in the decision procedure of joint activities’ planning. During the questionnaire survey, since it requires a certain number of researchers to illustrate the definition and implication of the influencing factors of decision making through semi-structured interviews, this research is supported by a small number or samples, which is a limitation, to some extent. Although the research about planning strategy has originally a small-sample characteristic [49], the data in this research represent a certain degree of value and meaning. However, in future studies, relative research works could optimize the sample selection by choosing and screening more facility samples. Furthermore, there is a certain degree of “negotiation” in some specific activities, which is also shown in the result of data analysis in this paper. Therefore, it is possible to further develop the research by optimizing the decision-making model with a distillation of “negotiation factors” in future studies.

## Figures and Tables

**Figure 1 ijerph-19-10424-f001:**
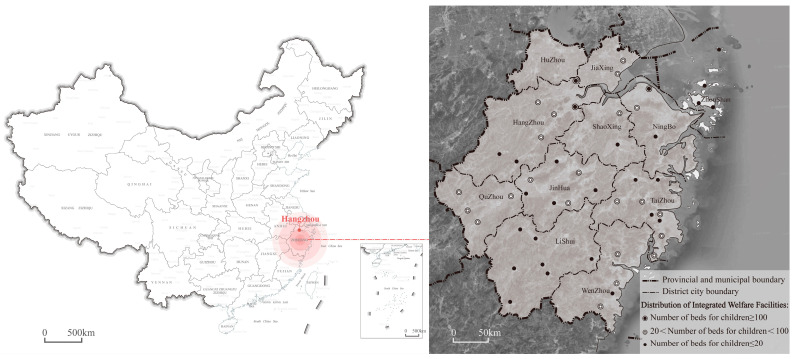
The distribution of integrated welfare facilities and child accommodation size in Zhejiang Province.

**Figure 2 ijerph-19-10424-f002:**
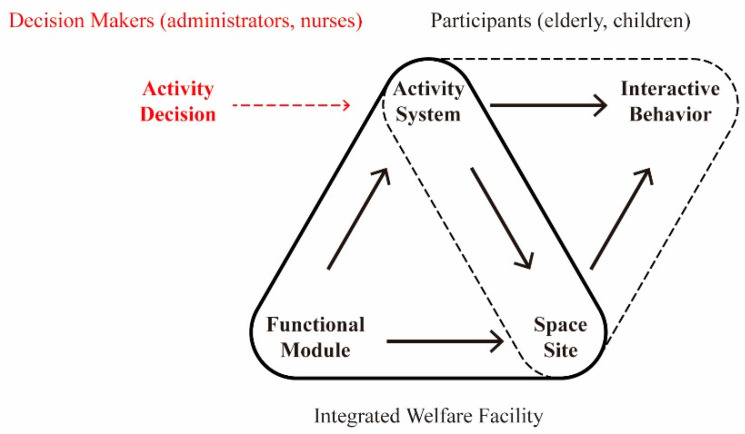
The “carrier-people” balance system of the joint activities of the elderly and children in welfare facilities.

**Figure 3 ijerph-19-10424-f003:**
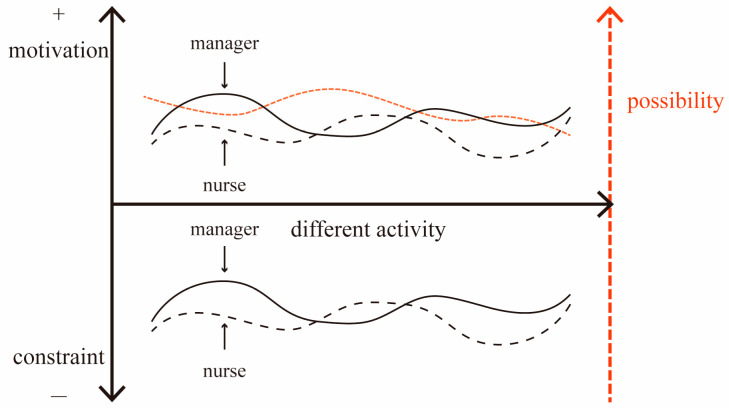
The basic structure of the decision-making model of the joint activity.

**Figure 4 ijerph-19-10424-f004:**
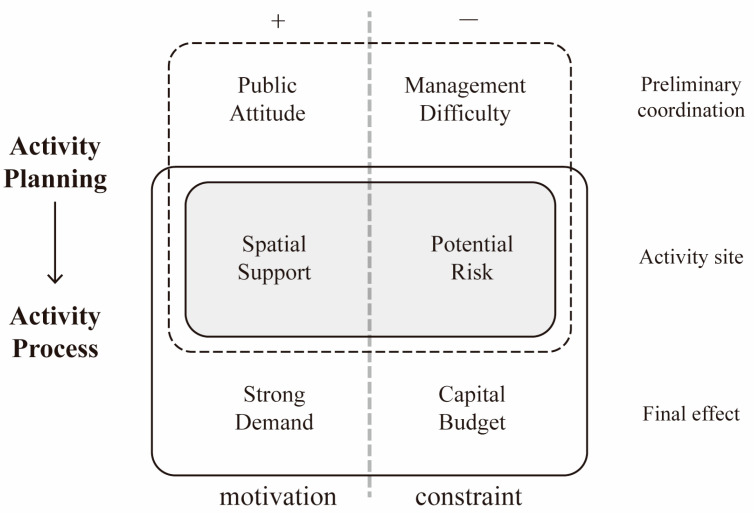
The level of “motivation–constraint” factors in the elderly–children activity.

**Figure 5 ijerph-19-10424-f005:**
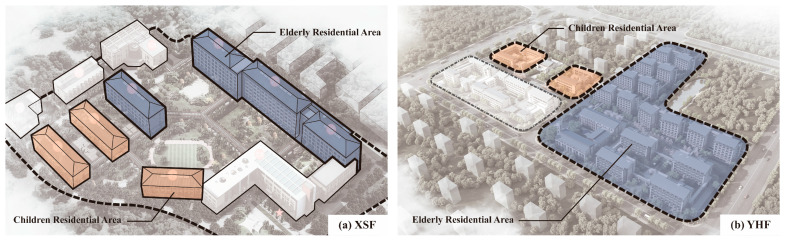
The residence distribution of the Xiao Shan Welfare Facility (XSF) and the Yu Hang Welfare Facility (YHF).

**Figure 6 ijerph-19-10424-f006:**
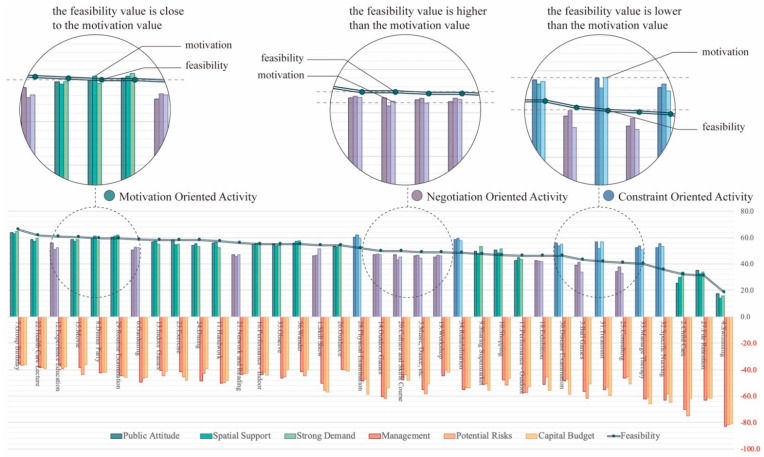
Deviation values of motivations, constraints and feasibilities obtained by different activities.

**Figure 7 ijerph-19-10424-f007:**
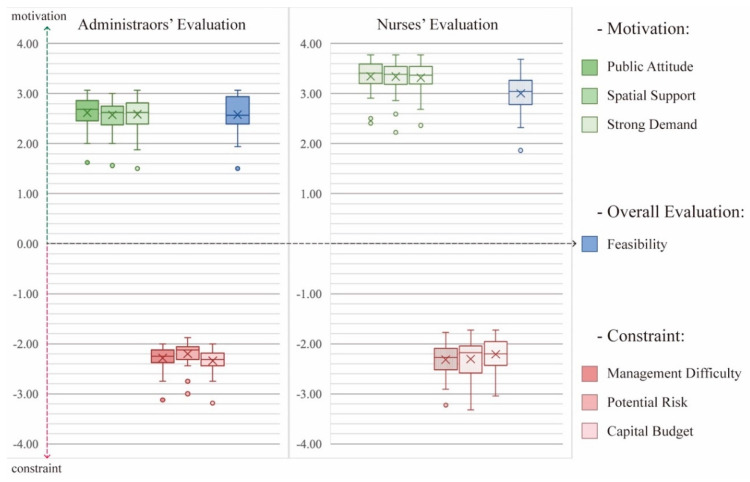
The evaluation range of administrators and nurses on the joint activities.

**Figure 8 ijerph-19-10424-f008:**
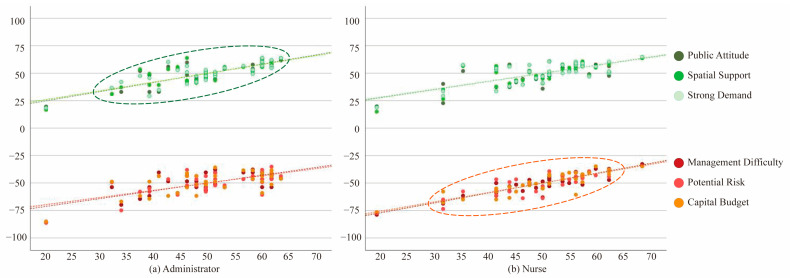
Three-dimensional scatter diagram of motivation, constraint values and feasibility of the joint activities of the elderly and children.

**Table 1 ijerph-19-10424-t001:** Basic information database of the joint activities of the elderly and children in integrated welfare facilities.

Activity Type	Activity Item	Source		Activity Content	Functional Space
		Recorded Before	Observed		
Skill Show	1. Skill Show		√	Children and elderly play together at a festival	Educational Activity Room
Volunteer	2. Child Care	√		Active elderly help look after children	
Hobby A	3. Ball Games	√	√	Billiards/badminton and other ball games	
	4. Swimming	√		Elderly and children share swimming pool	
	5. Music, Dance, etc.	√		Moderate intensity of old and young dance	
	6. Gardening	√	√	Co-plant and pick vegetables and flowers	
Settlement	7. Group Birthday	√		The elderly and children celebrate their birthday together	Residential Room
	8. Dinner Party	√	√	Gather together at festivals, such as Mid-Autumn Festival	
LifeSupport A	9. Sharing Supermarket	√	√	Scenarios that can teach children to learn the cash register	
	10. Shopping	√	√	The elderly and children buy daily necessities	
Sharing	11. Handwork	√	√	Children and elderly learn to cook food together	Educational Activity Room
	12. Experience Education	√		The elderly impart social experience to children	
Hobby B	13. Indoor Games	√	√	Interesting games, such as ferrules and chess	
	14. Outdoor Games	√	√	Interesting games, such as box folding competition	
Movieand Show	15. Movie	√	√	Children and elderly watch a film together	
	16. Performance—Indoor	√	√	Children and elderly enjoy performances, such as Yue opera	
	17. Performance—Outdoor	√	√	Elderly and children enjoy dog training together	
Retrieval	18. Exhibition	√		Children and elderly visit a theme exhibition	
	19. Workshop	√		Communication of cultural/life knowledge	
Skill Training	20. Culture and Skills Course	√		Learn a skill or culture together	
	21. Network and Reading	√		Read books, surf the internet	
Health control A	22. Health Care Lecture	√	√	Doctors popularize health care knowledge	
	23. Exercise	√	√	Daily morning exercises and daily exercise	
DailyAssistance	24. Dining	√	√	Eat together in the dining hall	Residential Room
Life Support B	25. Consulting	√		Professional provision consultation and guidance	
	26. Guidance	√		Daily life Q & A and help	
Health control B	27. File Retention	√		Deposit of personal belongings during activities	
	28. Physical Examination	√		Annual and quarterly physical examination	Health Care Room
	29. Routine Examination	√		Daily routine examination, such as blood pressure/temperature	
DiseaseTreatment	30. Disease Examination	√		Examination when the body is not in good condition	
	31. Treatment	√		Targeted treatment after disease diagnosis	
Recovery	32. Specific Nursing	√		Special care for teeth and other organs	
	33. Massage Therapy	√		Health care through physical massage	
	34. Rehabilitation	√		Recovery against some reversible damage	
Relaxation	35. Observe		√	The elderly watch and appreciate children’s activities	-
	36. Wander		√	Take a walk and relax in the park	

**Table 2 ijerph-19-10424-t002:** “Motivation–Constraint” element of joint activity in welfare facility.

Level	Motivation		Constraint	
Preliminary coordination	Public Attitude	The degree of support for the activities children and elderly participate in under the current public opinion environment.	Management Difficulty	The shortage degree of managers of the activity planned and held in the facility.
Activity site	Spatial Support	For integrated welfare facilities, the spatial scale, accessibility, openness and other conditions of space have to meet the requirements of activities.	Potential Risk	The degree of potential safety risks if the activity is held in the facility space.
Final effect	Strong Demand	The intensity of the preference or demand of the elderly and children to participate in the activity.	Capital Budget	The degree of the shortage of economic and financial funding for the activity planned and held in the facility.
Overall Feasibility Evaluation	Based on the above considerations, the extent to which the administrators and activity planners of welfare facilities evaluate whether the activity can be implemented.			

**Table 3 ijerph-19-10424-t003:** The basic Information of the Xiao Shan Welfare Facility (XSF) and the Yu Hang Welfare Facility (YHF).

Facility	Integrated Mode	Spatial Mode	Construction Scale	Number of Residents	Design Standards	Decision Makers	Elderly and Children	Activities Situation
XSF	Nursing homes and orphanages are constructed together as integrated welfare facilities	Joint-construction	Total: 41,783 m^2^	Elderly: 330 Children: 40	<Construction standard for integrated welfare facilities > (JB 179-2016), <Design code for buildings of elderly facilities > (GB 50 867- 2013)	Administrators:Dispatched by government departments; Nurses:Employed and trained according to employment standards	The elderly: From “Wubao” families and from normal families; Children: Received according to relevant reception standards	Already taking place, twice a month, through activity planning
YHF		Adjoining	The Elderly: 45,674 m^2^ Children: 7020 m^2^	Elderly: 300 Children: 50				Already taking place, once a month, through activity planning

As the elderly and children will move in and out, there may be a small difference in the “number of residents” living in facilities.

**Table 4 ijerph-19-10424-t004:** The basic information of the interviewees.

Interviewee’s Occupation/Position		Number of Interviewees		Percent Female (%)
Administrator	Director	XSF	1	100
		YHF	7	43
	Manager	XSF	4	100
		YHF	4	50
Nurse	Head Nurse	XSF	1	100
		YHF	2	100
	Nursing Staff	XSF	14	100
		YHF	5	60

**Table 5 ijerph-19-10424-t005:** Motivation-oriented activity database.

Activity	Public Attitude	Spatial Support	Strong Demand	Management	Potential Risk	Capital Budget	Feasibility
7. Group Birthday	** 63.763 **	** 62.726 **	** 64.850 **	−35.678	−36.586	−36.191	** 66.117 **
22. Health Care Lecture	**58.436**	**57.399**	**59.513**	−38.630	−38.381	−39.138	** 61.659 **
15. Movie	**58.436**	**57.399**	**58.623**	−38.630	−43.766	−36.191	** 60.174 **
8. Dinner Party	**59.324**	** 60.950 **	**57.734**	−42.565	−41.971	−42.086	**59.431**
29. Routine Examination	** 60.211 **	** 60.950 **	** 62.181 **	−44.533	−44.664	−46.015	**59.431**
13. Indoor Games	**56.660**	**57.399**	**55.065**	−40.598	−44.664	−41.103	**57.945**
23. Exercise	**58.436**	**54.735**	**55.065**	−41.581	−45.562	−47.980	**57.945**
24. Dining	**53.996**	**55.623**	**53.286**	−48.469	−42.869	−39.138	**57.945**
11. Handwork	**55.772**	**56.511**	**52.397**	**−50.437**	**−50.050**	−47.980	**57.202**
16. Performance—Indoor	**54.884**	**55.623**	**55.955**	−43.549	−41.971	−44.050	**54.973**
35. Observe	**54.884**	**53.847**	**54.176**	−46.501	−45.562	−40.121	**54.973**
36. Wander	**55.772**	**57.399**	**57.734**	−41.581	−44.664	−40.121	**54.973**
26. Guidance	**53.108**	**52.960**	**54.176**	−39.614	−40.176	−41.103	**54.230**
9. Sharing Supermarket	49.556	47.632	**53.286**	**−51.421**	**−50.947**	**−55.840**	47.544
10. Shopping	**50.444**	48.520	**51.507**	−47.485	**−51.845**	−46.998	46.801
17. Performance—Outdoor	42.452	44.969	43.502	**−57.325**	**−57.231**	**−52.893**	46.058
2. Child Care	25.581	29.875	33.717	** −70.117 **	** −75.183 **	** −61.735 **	31.943
27. File Retention	35.349	32.539	33.717	** −63.229 **	** −60.821 **	** −61.735 **	31.200
4. Swimming	17.590	13.894	15.927	** −82.909 **	** −81.467 **	** −81.385 **	18.571

Bold: | deviation value |≥ 50; Underline: | deviation value |≥ 60.

**Table 6 ijerph-19-10424-t006:** Negotiation-oriented activity database.

Activity	Public Attitude	Spatial Support	Strong Demand	Management	Potential Risk	Capital Budget	Feasibility
12. Experience Education	**55.772**	**51.184**	**52.397**	−36.662	−39.278	−38.156	** 60.917 **
6. Gardening	**50.444**	**52.960**	**52.397**	−49.453	−46.459	−46.015	**58.688**
21. Network and Reading	46.892	45.857	47.060	−43.549	−42.869	−43.068	**55.716**
1. Skill Show	46.004	46.745	**51.507**	**−50.437**	**−55.436**	**−56.823**	**54.230**
14. Outdoor Games	46.892	47.632	47.060	** −60.277 **	** −61.719 **	**−53.875**	49.773
20. Culture and Skills Course	46.892	43.193	45.281	−47.485	−43.766	−47.980	49.773
5. Music, Dance, etc.	46.004	46.745	44.391	**−55.357**	**−58.128**	**−50.928**	49.030
19. Workshop	45.116	46.745	46.170	−44.533	−40.176	−42.086	49.030
18. Exhibition	42.452	42.305	41.723	**−51.421**	−45.562	**−55.840**	46.058
3. Ball Games	38.901	41.417	33.717	**−56.341**	** −61.719 **	**−50.928**	43.087
25. Consulting	34.461	37.866	32.828	−46.501	−45.562	**−50.928**	40.858

Bold: | deviation value |≥ 50; Underline: | deviation value |≥ 60.

**Table 7 ijerph-19-10424-t007:** Constraint-oriented activity database.

Activity	Public Attitude	Spatial Support	Strong Demand	Management	Potential Risk	Capital Budget	Feasibility
28. Physical Examination	** 60.211 **	** 61.838 **	**59.513**	−48.469	−47.357	**−58.788**	**52.002**
34. Rehabilitation	**58.436**	**59.175**	**57.734**	−55.357	−53.640	**−53.875**	48.287
30. Disease Examination	**55.772**	**53.847**	**55.065**	−48.469	−47.357	**−58.788**	46.058
31. Treatment	**56.660**	**52.072**	**56.844**	**−55.357**	**−53.640**	**−59.770**	41.601
33. Massage Therapy	**52.220**	**53.847**	**50.618**	** −62.245 **	** −60.821 **	** −65.665 **	40.115
32. Specific Nursing	**52.220**	**55.623**	**53.286**	** −63.229 **	**−58.128**	** −64.683 **	35.658

Bold: | deviation value |≥ 50; Underline: | deviation value |≥ 60.

**Table 8 ijerph-19-10424-t008:** Different evaluation of activities from administrators and nurses.

Activity	Feasibility		Motivation						Constraint					
			Public Attitude		Spatial Support		Strong Demand		Management Difficulty		Potential Risk		Capital Budget	
	A	N	A	N	A	N	A	N	A	N	A	N	A	N
Skill show	●	-	●	-	●	-	●	-	⊚	-	⊚	⊚	⊚	⊚
Culture and Skill Course		●	-	-	-	●	-	●	-	-	-	-	-	⊚
Workshop		●	-	-	-	-	-	●	-	-	-	-	-	-
Physical Examination		●	●	●	●	●	●	●	⊚	-	-	-	⊚	⊚
Rehabilitation		●	●	●	●	●	●	●	⊚	⊚	⊚	-	⊚	-

●: deviation value ≥ 50; -: −50 < deviation value < 50; ⊚ deviation value ≤ −50.

**Table 9 ijerph-19-10424-t009:** Classification and assessment criteria for the joint activities.

Activity Type	Assessment Criteria		Active Status
A Should occur	Already occurred	High feasibility	Activities proven to be suitable for integrated welfare facilities, which should maintain the current condition.
B Occurred but should be improved		Low feasibility	Activities that should improve the safety and raise the budget.
C Potentially could occur	Not occurred yet	High feasibility	Activity may need space and other resources that are not yet available.
D Hard to occur		Low feasibility	Activities that proved inappropriate for integrated welfare facilities.

**Table 10 ijerph-19-10424-t010:** Items and corresponding status of activities that “Should Occur”.

Activity	Feasibility		Assessment Accuracy	
	A: Administrator	B: Nurse	A: Administrator	B: Nurse
6. Gardening	53.031	**62.178**	1	1
7. Group Birthday	**61.691**	**68.301**	1	1
8. Dinner Party	**61.691**	57.280	1	1
11. Handwork	**61.691**	53.606	1	1
12. Experience Education	58.227	**62.178**	1	1
13. Indoor Games	59.959	56.055	1	1
15. Movie	59.959	59.729	1	1
16. Performance—Indoor	59.959	51.157	1	1
23. Exercise	58.227	57.280	1	1
35. Observe	51.299	57.280	1	1
36. Wander	56.495	53.606	1	1

Bold: deviation value ≥ 60; Accuracy = 1: the judgment is consistent with the current situation of activity; Accuracy = 0: the judgment is inconsistent with the current situation of activity.

**Table 11 ijerph-19-10424-t011:** Items and corresponding status of activities that “Occurred but Should be Improved”.

Activity	Feasibility		Assessment Accuracy	
	A: Administrator	B: Nurse	A: Administrator	B: Nurse
1. Skill Show	59.959	49.932	1	0
9. Sharing Supermarket	47.835	47.483	0	0
10. Shopping	46.103	47.483	0	0
14. Outdoor Games	49.567	49.932	0	0
17. Performance—Outdoor	49.567	43.809	0	0
19. Workshop	46.103	51.157	0	1
28. Physical Examination	46.103	56.055	0	1
34. Rehabilitation	44.371	51.157	0	1

Bold: deviation value ≥ 60; Accuracy = 1: the judgment is consistent with the current situation of activity; Accuracy = 0: the judgment is inconsistent with the current situation of activity.

**Table 12 ijerph-19-10424-t012:** Items and corresponding status of activities that “Potentially Could Occur”.

Activity	Feasibility		Assessment Accuracy	
	A: Administrator	B: Nurse	A: Administrator	B: Nurse
20. Culture and Skills Course	47.835	51.157	1	0
21. Network and Reading	51.299	58.504	0	0
22. Health Care Lecture	59.959	**62.178**	0	0
24. Dining	**61.691**	54.830	0	0
26. Guidance	51.299	56.055	0	0
29. Routine Examination	**63.423**	56.055	0	0

Bold: deviation value ≥ 60; Accuracy = 1: the judgment is consistent with the current situation of activity; Accuracy = 0: the judgment is inconsistent with the current situation of activity.

**Table 13 ijerph-19-10424-t013:** Items and corresponding status of activities that are “Hard to Occur”.

Activity	Feasibility		Assessment Accuracy	
	A: Administrator	B: Nurse	A: Administrator	B: Nurse
2. Child Care	33.979	31.563	1	1
3. Ball Games	39.175	46.258	1	1
4. Swimming	20.123	19.317	1	1
5. Music, Dance, etc.	49.567	48.707	1	1
18. Exhibition	47.835	45.034	1	1
25. Consulting	40.907	41.360	1	1
27. File Retention	32.247	31.563	1	1
30. Disease Examination	49.567	43.809	1	1
31. Treatment	42.639	41.360	1	1
32. Specific Nursing	37.443	35.237	1	1
33. Massage Therapy	39.175	41.360	1	1

Bold: deviation value ≥ 60; Accuracy = 1: the judgment is consistent with the current situation of activity; Accuracy = 0: the judgment is inconsistent with the current situation of activity.

**Table 14 ijerph-19-10424-t014:** Activity types and feasibility classified in different service modules.

Service	Type	Activity	Feasibility		Present Situation		Assessment Accuracy			Type
			A: Administrator	B: Nurse	Already Occurred	Not Occurred Yet	A: Administrator	B: Nurse		
**Daily Support**	DailyAssistance	7. Group Birthday	**61.691**	**68.301**	√		1		1	A
		8. Dinner Party	**61.691**	57.280	√		1		1	A
		24. Dining	**61.691**	54.830		√	0		0	C
	Additional help	26. Guidance	51.299	56.055		√	0		0	C
		9. Sharing Supermarket	47.835	47.483	√		0		0	B
		10. Shopping	46.103	47.483	√		0		0	B
		25. Consulting	40.907	41.360		√	1		1	D
**Medical Care**	Health control	29. Routine Examination	**63.423**	56.055		√	0		0	C
		23. Exercise	58.227	57.280	√		1		1	A
		22. Health Care Lecture	59.959	**62.178**		√	0		0	C
		28. Physical Examination	46.103	56.055	√		0		1	B
		27. File Retention	32.247	31.563		√	1		1	D
	DiseaseTreatment	30. Disease Examination	49.567	43.809		√	1		1	D
		31. Treatment	42.639	41.360		√	1		1	D
	Recovery	34. Rehabilitation	44.371	51.157	√		0		1	B
		33. Massage Therapy	39.175	41.360		√	1		1	D
		32. Specific Nursing	37.443	35.237		√	1		1	D
**Education and Culture**	Skill Training	21. Network and Reading	51.299	58.504		√	0		0	C
		20. Culture and Skills Course	47.835	51.157		√	1		0	C
	Retrieval	18. Exhibition	47.835	45.034		√	1		1	D
		19. Workshop	46.103	51.157	√		0		1	B
**Social Practice**	Skill Show	1. Skill Show	59.959	49.932	√		1		0	B
	Sharing	11. Handwork	**61.691**	53.606	√		1		1	A
		12. Experience Education	58.227	**62.178**	√		1		1	A
	Volunteer	2. Child Care	33.979	31.563		√	1		1	D
**Entertainment**	Movie and Show	15. Movie	59.959	59.729	√		1		1	A
		16. Performance—Indoor	59.959	51.157	√		1		1	A
		17. Performance—Outdoor	49.567	43.809	√		0		0	B
	Hobby	13. Indoor Games	59.959	56.055	√		1		1	A
		6. Gardening	53.031	**62.178**	√		1		1	A
		14. Outdoor Games	49.567	49.932	√		0		0	B
		5. Music, Dance, etc.	49.567	48.707		√	1		1	D
		3. Ball Games	39.175	46.258		√	1		1	D
		4. Swimming	20.123	19.317		√	1		1	D
	Relaxation	36. Wander	56.495	53.606	√		1		1	A
		35. Observe	51.299	57.280	√		1		1	A

Bold: deviation value ≥ 60; Accuracy = 1: the judgment is consistent with the current situation of activity; Accuracy = 0: the judgment is inconsistent with the current situation of activity.

## Data Availability

Data supporting the reported results can be found at https://pan.zju.edu.cn/share/653b4bb021e9761a5b3feeda45 (accessed on 18 July 2022).

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
