# Peer review of "Decision-Making Mechanism of Joint Activities for the Elderly and Children in Integrated Welfare Facilities: A Discussion Based on “Motivation–Constraint” Interaction Model"

_ijerph, 2022, doi:10.3390/ijerph191610424_

Round 1

Reviewer 1 Report

Dear Authors,

This paper addresses an interesting topic, however, I would recommend several modifications before considering its publication. Below these are some suggestions for You:

  1. The overall, type, title and construction of the paper:

    1. I do not understand what type of article it is. It should be readable at first glance. 

    2. Reconstruct the manuscript or rename titles of sections, according to ‘Instructions for Authors’. Why is there a ‘literature review’ in the middle of nowhere? Two sections forward the section ‘discussion’ occurs. Or clarify your aim, or change your title (like: literature review and study). Maybe the manuscript should be divided into two smaller parts (two articles).Reading this article is very overwhelming

  2. Literature review:

    1. There are two tables 2 - one of the (line 202) is not mentioned in the text. Second, where is table 1?

  3. Material and methods: 

    1. Any eligibility criteria? 

    2. Table 3 - X i Y are very similar letters, so maybe you can rename examined facilities; another issue, the table should be reconstructed to be more readable. You may also mention the number of residence in the table. 

    3. Why 17 questions from the Y facility (line 309) were invalid? And what about the X facility?

  4. Results:

    1. Tables 9-12 are not mentioned in the main text.

    2. Figure 6, 7, 8 not readable (too small).

    3. Figure 7 - state the (a) and (b) as well as colours meaning. 

    4. There is any comparison between the two facilities? 

  5. Discussion: 

    1. Rename the title (‘discussion’ is enough)

    2. Discuss limitations of the study

  6. References:

    1. Please, search for some new references. 

Best regards and good luck

Author Response

Dear reviewer:
Many thanks for your comments. The manuscript has been revised accordingly. The details are listed as below (please find the figure and table in the attachment):

1. The overall, type, title and construction of the paper:
a. I do not understand what type of article it is. It should be readable at first glance.
Response: Sorry for causing the difficulties of understanding the article. Considering this problem you mentioned, we rethought and reorganized the manuscript by summarizing 4 main problems. Firstly, the original title may not clearly present the main idea of the study. Secondly, the abstract of this paper lacks a description of the result. Thirdly, as you pointed out in the following, the structure of this paper is too complicated. Fourthly, the literature review and data analysis of this study are similar in length, which may easily lead to questions about the type of articles for readers.
Firstly, we replaced the original title “‘Motivation-Constraint’ Decision-Making Mechanism of Joint Activities of the Elderly and Children in Integrated Welfare Facilities” with “Decision Making Mechanism of Joint Activities for the Elderly and Children in Integrated Welfare Facilities: A Discussion based on ‘Motivation-Constraint’ Interaction Model”. In this new title, we illustrate the research objects, objectives, and methods. Additionally, our research object is “Joint Activities for the Elderly and Children” and our research objective is to figure out a “Decision-Making Mechanism” which is the key point of this study. And “‘Motivation-Constraint’ Interaction Model” is the core model and method to achieve this objective.

Secondly, as for the abstract, we add the description of the results in the abstract part, which is updated from line 13 to 28:

In China, joint activities for the elderly and children in integrated welfare facilities lack systematic decision procedures. By learning from the “leisure constraint” theory, the study put forward 6 influencing indicators of motivation and constraint in the aspects of preliminary coordination, activity space and effect. By semi-structured interviews and questionnaire surveys analyzed by deviation value computation, the study analyzes the evaluation value of influencing factors in decision procedure of potential activity cases, where administrators and nurses act as two decision-makers. Further, it discusses the decision making mechanism based on “motivation-constraint” interaction model. Firstly, it analyses the dominated forces in decision procedure, which are “motivation oriented”, “negotiation oriented”, and “constraint oriented”. Secondly, it reveals that administrators and nurses as two decision-makers tend to give positive motivation evaluations and deliberative constraints evaluations respectively. And it analyzes the decision procedures of activities with distinct feasibility differentiation. Thirdly, it positions the levels of occurrence potential as “should occur”, “occurred but should be improved”, “potentially could occur”, and “hard to occur”. Eventually, it analyzes the requirements and potential for joint activities under different service modules, which provides a theoretical foundation for the systematic planning and development of the joint activities.

Thirdly, to avoid difficulties in understanding, the chapters of literature review and research methods are merged into "Literature Review and Methods" with appropriate deletion and optimization. The purpose of this chapter is to explore methods to find a research theoretical model fitting this research, which can be used as the analysis support for the results in Chapter 3 and the discussion about “decision-making tendency and development potential of the activities of different service modules” in Chapter 4.

Fourthly, as for the problem that the literature review and data analysis of this study are similar in length, due to the lack of basic research in relevant fields, there are few pieces of research on activity decision-making methods for joint activities. In order to enhance the scientific nature of this research, the literature review part of this paper should describe the logical nodes including "decision theory - activity type - model index - facility sample". In the meantime, we simplify the theoretical discussion and highlight the key points of the paper to improve its readability.

b. Reconstruct the manuscript or rename titles of sections, according to ‘Instructions for Authors’. Why is there a ‘literature review’ in the middle of nowhere? Two sections forward the section ‘discussion’ occurs. Or clarify your aim, or change your title (like: literature review and study).

Response: Thanks for your suggestion, we checked “Instructions for Authors” again. The Instructions mentioned that "Research manuscript sections" should include Introduction, Materials and Methods, Results, Discussion, and Conclusions. According to this requirement, we merged the original “2. Literature Review” and “3. Research Methods” into “2. Literature Review and Methods”. The secondary titles are: “2.1. The Introduction of The Leisure Constraint Theory”, “2.2. Sample Extraction of the Joint Activities of the Elderly and Children”, “2.3 The Decision-making Framework Development of the Joint Activities for the Elderly and Children in Integrated Welfare Facilities”, and “2.4 Data Acquisition and Analysis Methods”. Besides, this research also improves the content of Chapter 2. For example, in “2.1 The Introduction of the Leisure Constraint Theory”, the application scope of leisure constraint theory is supplemented (line 151 to 161) and distills the description of leisure constraint negotiation model (Line 176). In “2.2 Sample Extraction of the Joint Activities of the Elderly and Children”, the purpose to discuss the activities from other countries is also summarized (line 181), which is conducive to the coherence and readability of the updated manuscript.

Maybe the manuscript should be divided into two smaller parts (two articles). Reading this article is very overwhelming

Response: We carefully reconsidered the feasibility of dividing the manuscript into two parts: One of which could be a discussion on the mechanism of the joint activities in China’s integrated welfare facilities, and the other part could be an empirical study of decision making results of the joint activities based on “motivation-constraint” interaction model. Considering the length of the manuscript, it is feasible to construct profound short-length essays with more factors on specific entry points. However, for some pieces of “pioneering research” without sufficient previous research foundation, it is necessary to construct a certain length of the manuscript to interpret the cause and effect as well as the multi-node logic chain. Since this manuscript is a “pioneering research” without sufficient research foundation, one of its contributions is to address the issue of the joint activities in China’s integrated welfare facilities, which illustrates the tripartite logical relationship, which is “occurrence mechanism of the joint activities - choosing of the joint activities decision making models- discussion and analysis of the joint activities system” and constructs a comprehensive reasoning framework. Therefore, we prefer to keep the completeness of the manuscript instead of dividing it into two parts. However, inspired by your suggestion, we plan to conduct further short-length systematic research on both occurrence mechanism and activities decision making process in the future. It will be based on a foundation of adequate literature review which includes more possible influencing factors such as negotiation and so on besides motivation and constraints.

2. Literature review:

a. There are two tables 2 - one of the (line 202) is not mentioned in the text. Second, where is table 1?

Response: We do apologize for this omission, and we have modified the serial number of Table 1 in the manuscript (line 218), and mentioned it in line 217.

3. Material and methods:
a. Any eligibility criteria?

Response: In our previous version, we lacked a description of the eligibility criteria for facility selection, which leads to the problem of unclear characteristic facilities.
In “1. Introduction”, this manuscript first supplements the phenomenon of integrated welfare facilities and the characteristics of participants (line 52 to 73):
In the aspect of space condition of welfare facilities built in integrated mode, although there is a “joint-construction” type that different facilities built in one single park and the other type called “adjoining” that different facilities built in separate parks that adjoin closely, actually in their nature, under the positive condition of joint management and adjacent location, the intimate living spaces of the elderly and children will stimulate the generation of joint activities between these two groups. By observation, these joint activities do not mean the elderly and children long-termly live together in one space, however, most of time, these two age groups of people are taken care of in their own functional spaces and carry on basic daily activities. Besides, planned by facility staff in a certain frequency, these joint activities are held randomly by borrowing some spaces in the elderly or children’s own functional spaces. Normally, orphans attend activities under facility staff’s guidance and the elderly attend activities on their own. Additionally, since most of the joint activities have strong communicative attributes, most of the elderly who are willing to attend the joint activities have good physical and mental conditions as well as a high sense of social identity and family cognition. Compared with the basic elderly security provided for high-risk families (“Wubao” families) by the government before, in recent years, the private capital has been gradually involved in the operation of these welfare facilities. Further, the scope of services of these facilities relatively expanded that they also accepted the elderly from normal families under the support of the government, improving the average mental and physical health quality of the elderly [3]. Also, the participation rate and experience of these joint activities have been improved which also benefit the sustainability of the operation of these activities.
3. Qiu, Z.; Luo, W.J.; Wang, Z. The Methods of Space Design in Integrated Care Welfare Facility Based on "Social Support" Theory. Architectural Journal. 2021, S2, 47-52. (In Chinses)

Based on the current situation, the characteristics of facilities vary greatly in different cities because of different economic and geographic background. If we choose facilities from different cities as our samples, it will lead to the problem of an inconsistent decision-making background of joint activities. Therefore, we only selected two typical facilities in the economically developed Hangzhou for our investigation. In terms of management and operation, the two facilities both are managed by the Elderly Care Service Department , as well as the Juvenile Protection and Child Welfare Division of Zhejiang Civil Affairs Bureau. Besides, some of the administrators in the facilities are appointed by the government and the nurses are employed in accordance with the national professional skills standards for nurses . In addition to the elderly from “Wubao” families without family members, more than half of the elderly from normal families have been accepted to the facilities, and have a strong willingness to participate in activities. Moreover, the children are mainly supervised and taken care of by nurses in both of the facilities. Although these two facilities have certain differences in scale and construction form, they share the same management mode and similar characteristic of group of people inside the facilities. Further, they both face same activity decision-making problems. Therefore, we selected them as two typical facilities for investigation in this study. In the updated 2.4, we supplemented the selection criteria of integrated welfare facilities (line 305 to 316):

To avoid the differentiation of the decision making bodies and other decision making factors of the elderly and children activities caused by the different regions of welfare facilities, this research chooses two typical integrated welfare facilities in Hangzhou (Figure 5) which are Xiao Shan Welfare Facility (XSF) and Yu Hang Welfare Facility (YHF). Although in the aspect of space condition, these two integrated welfare facilities respectively belong to the “joint-construction” type and the “adjoining” type and vary in their sizes, they both obey strictly to the integrated welfare facility architecture design standard in administrative district level. Additionally, after construction, both of them are in the charge of the same government sector which is the Civil Administration Office in Zhejiang Province. Also, there is a certain degree of similarity of people groups including the elderly, children, administrators, and nurses. Furthermore, both of them have a certain amount of joint activities that happened inside in a certain frequency (Table 3 in updated manuscript).

b. Table 3 - X i Y are very similar letters, so maybe you can rename examined facilities; another issue, the table should be reconstructed to be more readable. You may also mention the number of residence in the table.

Response: In terms of the name of the facilities, to avoid this misunderstanding, we changed the name of the facilities in the updated manuscript. Originally, X facility represented Xiao Shan Integrated Welfare Facility and Y facility represented Yu Hang integrated welfare facility, we will use XSF and YHF as their name. Additionally, we have reconstructed the form and include the number of the elderly and children residing there in table3 (line 649).

c. Why 17 questions from the Y facility (line 309) were invalid? And what about the X facility?

Response: The criteria for invalid questionnaires have been briefly explained in Chapter 2.4 of the main text (line 330 to 332):
By qualitative examination of data, 17 invalid questionnaire surveys are being eliminated, which contain mistakes such as overmuch mechanically repeated numbers or obvious inconsistency.

In addition, the questionnaire survey in this manuscript conducts the semi-structured interview, so theoretically the rate of the invalid questionnaires will be relatively low, but this method also relies on the cooperation and support of respondents. The questionnaire survey implemented in the XSF was relatively successful, and there was no invalid questionnaire. But in the process of the YHF questionnaire survey, we happened to meet an emergency event. So the quality of the questionnaires from several nurses was affected. Therefore, there are 17 invalid questionnaires which contain mistakes such as overmuch mechanically repeated numbers or obvious inconsistency. As a result, we eliminated these 17 questionnaires to represent the process of doing this survey and to ensure the reliability of the data.

4. Results:

a. Tables 9-12 are not mentioned in the main text.
b. Figure 6, 7, 8 not readable (too small).
c. Figure 7 - state the (a) and (b) as well as colours meaning.

Response: We apologize for the deficiencies in the tables and figures. In the updated manuscript, we mention Table 9-12 in the main text, enlarge and make Figure 6,7,8 more readable. Also, we add legends in the Figure 7.
Besides, the differentiation of different types of activities in Figure 6 are relatively microscopic compared with the overall picture. To enhance the readability of this paper, we added enlarged pictures of different types of activities to make the differentiation between motivation and feasibility more distinctive. It can be observed from the figure that the motivation value of activities is very close to the value of feasibility, which means the motivation of activity plays a major role. However, the motivation of the second type of activity is significantly lower than the feasibility level, indicating that there is a certain negotiation process to improve the feasibility of activities in this type of activity. The motivation of the third type of activity is significantly higher than the feasibility, indicating that the restriction of the activity is difficult to overcome, which reduces the possibility of the activity being implemented.

This differentiation is used as the evaluation and classification of “motivation oriented”, “negotiation oriented” and “constraint oriented” activities.

d. There is any comparison between the two facilities?

Response: According to your previous question about “eligibility criteria” in “3. Material and methods”, this manuscript supplements the criteria and comparison for sample selection based on the updated definition of integrated welfare facilities (1. Introduction, line 91 to 112), which is showed from line 632 to line 643, and table 3 of “2.4 Data Acquisition and Analysis Methods”. The details of the modification are shown in the reply to your “question a” in “3. Material and methods”.
In addition, we also supplement the data processing methods for the two facilities in this chapter (line 325 to 329):

Although there is a certain amount of facilities’ staffs, only a few administrators and nurses could anticipate in the decision making process. Based on the comparison and analysis on both two facilities which reduce the decision making difference caused by facilities’ distinction, this research adds together the sample numbers in both facilities to keep a certain amount of samples.

That means, although these two facilities have differences in scale and construction form, they both obey the same management mode and characteristics of groups of people inside. Moreover, both of them have generated joint activities at a certain frequency. As a result, XSF and YHF are considered the same kind of samples in this study, and their data is considered as a whole group in the following calculation and analysis. Therefore, we did not compare the evaluation results between the two facilities in this manuscript.

5. Discussion:

a. Rename the title (‘discussion’ is enough).
b. Discuss limitations of the study

Response: 
Thanks for your advice, and we simplified the title you mentioned to “discussion”.
According to question b, this research prefers to be recognized as “pioneering research”, and to some extent, it does lack sufficient research conditions and opportunities, leading to the limitations of the study. In Chapter 5 of the updated manuscript, we conclude two limitations. One is the problem of the small sample, and the other is the problem of a relatively simple selection of influencing factors of the model:

However, although the welfare development progress in China has been acknowledged throughout the world, as a developing country, it still needs further model clarification, mechanism development, space construction, and evaluation of management and operation. Therefore, it still lacks mature conditions and opportunities for discussion on the elderly and children’s joint activities in welfare facilities. And that is the main reason why this research prefers to be recognized as “pioneering research” and it shows research limitations in several aspects. For example, although there is a certain amount of staff working in integrated welfare facilities, only a few of them could participate in the decision procedure of joint activities planning. During the questionnaire survey, since it requires a certain amount of researchers to illustrate the definition and implication of influencing factors of decision making through semi-structured interviews, this research has small sample support which is a limitation to some extent. Although planning research has originally a small-sample characteristic [49], the data in this research represents a certain degree of value and meaning. But in future studies, relative researches could optimize the sample selection by choosing and screening more facility samples. (line 624 to 638)

Furthermore, there is a certain degree of “negotiation” in some specific activities which is also shown in the result of data analysis in this paper. Therefore, it is possible to further develop the research by optimizing the decision making model with a distillation of “negotiation factors” in future studies. (line 638 to 642)

49. Tang, W,F.; Big Data and Small Data: A Discussion on social science research methods. Journal of Sun Yat-sen University (Social Science Edition). 2015, 55(6), 141-146.DOI:10.13471/j.cnki.jsysusse.2015.06.014. (In Chinses)

6. References:

a. Please, search for some new references.

Response: According to your suggestions, we add some new references about the leisure constraint theory to explain the application scope of leisure constraint theory more clearly (line 695 to 704), and about architectural programming with a characteristic of “small sample survey” (line 757).

19. Tom, H,; Edgar, L.J.; Simon, H.; Gordon, W. Leisure Constraint Theory and Sport Tourism. Sport in Society. 2005, 8(2), 142-163. 10.1080/17430430500087435
20. Hinch, T.D.; Jackson, E.L.; Leisure Constraints Research: Its Value as a Framework for Understanding Tourism Seasonability. Current Issues in Tourism. 1995, 27(1), 87-106. 10.1080/00222216.1995.11969978
21. Jackson, E.L.; Rucks, V.C.; Negotiation of Leisure Constraints by Junior-High and High-School Students: An Exploratory Study. Journal of Leisure Research. 1995, 27(1), 85-105. 10.1080/00222216.1995.11969978
22. McGuire, F.A.; Dottavio, D.; O’Leary, J.T. Constraints to participation in outdoor recreation across the life span: A nation-wide study of limitors and prohibitors. The Gerontologist. 1986, 26(5), 538-544. https://doi.org/10.1093/geront/26.5.538
23. Jackson, E.L.; Witt, P.A. Change and Stability in Leisure Constraints: A Comparison of Two Surveys Conducted Four Years Apart. Journal of Leisure Research. 1994, 26(4), 322-336. 10.1080/00222216.1994.11969965
49. Tang, W,F,; Big Data and Small Data: A Discussion on social science research methods. Journal of Sun Yat-sen University (Social Science Edition). 2015, 55(6), 141-146.DOI:10.13471/j.cnki.jsysusse.2015.06.014. (In Chinses)

The above has been revised responses to the manuscript. Your careful review has provided more help to our study clearer and more comprehensive. Thanks again for your advice!

Reviewer 2 Report

Merits of the paper

This research distilled the experience of the joint activities of the elderly and children in different countries. And based on the occurrence mechanism, the paper reveals a decision-making mechanism in the integrated welfare facilities in China, screening appropriate activity types for integrated welfare facilities. The paper also explains the requirements for holding various activities and the corresponding methods for tackling difficulties. The findings provide a theoretical basis for further program planning and architecture design of the integrated welfare facilities.

The paper screens the activities according to the cultural and social context of China at the first step and categorized them into 5 service supporting modules which are daily support, medical care, education&culture, social practice, and entertainment and their corresponding functions and spaces. And then a basic information database of the joint activities of the elderly and children in integrated welfare facilities is formed.

The research distills 3 motivation factors that are public attitude, spatial sharing, and strong demand as well as 3 constraints factors that are management difficulty, potential risk, and capital budget, and also evaluate the feasibility of each activity. Based on this, the paper conducts “motivation-constraints” models of the decision-making process.

The paper brings forward a basic structure decision-making model consisted of activity cases, decision-makers, motivations and constraints. The research investigates the level of motivation and constraints according to the evaluation of different sub-items and explores the relationship between different influencing factors and the feasibility. Based on this, the research compares and studies the evaluation difference between the administrators and nurses.

This study brings forward three hypotheses about the decision-making mechanism of the joint activities based on the research above.

Hypothesis 1: There is differentiation in the decision-making mechanisms of the elderly-children activities in the "Daily Support, Medical Care, Education & Culture, Social Practice and Entertainment" service module in the integrated welfare facilities.

Hypothesis 2: The degree of each motivation and constraint factor influencing the joint activities of the elderly and children in integrated welfare facilities varies from administrators to nurses.

Hypothesis 3: The evaluation standards and the validity of evaluation varies from administrators to nurses.

Because the joint activities are generally supported by the operation groups in welfare facilities and the decision-making process is influenced by positive and negative factors, this paper conducts leisure constraints models in tourism. It quantifies and discusses the motivation and constraints behind the decision-making process of the joint activities of the elderly and children. It collects the joint activities cases and builds a potential database of them screened by the decision-making model. And in this decision-making model of the joint activities of the elderly and children, both management administrators representing the perspective of the government and supervisors together with nurses with perspectives from users and real conditions act as an influential part of the decision-making process and they are regarded as the main body of the decision-makers.

Demerits of the paper

1. The research has very small data support - only 38 valid questionnaires were collected and analyzed.

2. Some important aspects of the integrated welfare facilities are not studied in the paper.

It will be good to include at least some comments for the next problems.

- The “leisure constraint” theory in tourism studies is aimed to support decision-making for temporary activities during given organized tourist’s trip or group relaxing. The integrated welfare facilities suppose long term joining the elderly and children. This main difference is very important for decision-makers due to some specific aspect of the second case. For example, only during a long interaction some conflicts between participants may grow up to irresolvable withstanding.

- Another problem is the mental deviations in very high ages which may cause not normal influence on the children.

- Joining the elderly and children is very dangerous from epidemic point of view. The social recursion of distributing the infection, for instance COVID-19, is “elderly-children-elderly”.

Author Response

Dear reviewer:
Many thanks for your comments. The manuscript has been revised accordingly. The details are listed as below:

1. The research has very small data support - only 38 valid questionnaires were collected and analyzed.

Response: Although the jointing of the elderly and children in China’s public welfare facilities is a byproduct generated during the process of the co-construction of nursing homes and orphanages due to land resources limitation, it has a great positive impact on improving the service quality for the elderly and caring for children. However, there is very limited fundamental research in the relevant field. Further, the joint activities of the elderly and children still remain in an exploration period. Therefore, it is necessary to clarify various types of unsettled issues on this topic, such as what content the elderly and children joint activity system should consist of in China’s integrated welfare facility, whether the elderly and children are willing to attend these activities, how to do architecture design for the jointing of the elderly and children activities, and so on. Therefore, this research could be recognized as “pioneering research”. Correspondingly, it still lacks mature conditions and opportunities to do research to some extent, which leads to limitations in this research, especially in the aspect of the sample amount. For this point, we have a supplementary explanation for this limitation in the Conclusion part of the manuscript from line 630 to line 638:

…… although there is a certain amount of staff working in integrated welfare facilities, only a few of them could participate in the decision procedure of joint activities planning. During the questionnaire survey, since it requires a certain amount of researchers to illustrate the definition and implication of influencing factors of decision making through semi-structured interviews, this research has small sample support which is a limitation to some extent. Although planning research has originally a small-sample characteristic [49], the data in this research represents a certain degree of value and meaning. But in future studies, relative researches could optimize the sample selection by choosing and screening more facility samples.
49. Tang, W,F.; Big Data and Small Data: A Discussion on social science research methods. Journal of Sun Yat-sen University (Social Science Edition). 2015, 55(6), 141-146.DOI:10.13471/j.cnki.jsysusse.2015.06.014. (In Chinses)

2. Some important aspects of the integrated welfare facilities are not studied in the paper.
It will be good to include at least some comments for the next problems.
- The “leisure constraint” theory in tourism studies is aimed to support decision-making for temporary activities during given organized tourist’s trip or group relaxing. The integrated welfare facilities suppose long term joining the elderly and children. This main difference is very important for decision-makers due to some specific aspect of the second case. For example, only during a long interaction some conflicts between participants may grow up to irresolvable withstanding.

Response: As you mentioned, the most typical theory in tourism studies is aimed to figure out the influencing factors of temporary activities during a given organized tourist trip. Besides, we also found some research that aim at the constraints for single or multiple participants who participate in continual activities for a long period time. The later research mode is appropriate for the joint activities in integrated welfare facilities which are in long-term operation.
For example, Change and Stability in Leisure Constraints- a Comparison of Two Surveys Conducted Four Years Apart discusses the leisure constraint in the same population by two surveys conducted four years apart (1988 and 1992). And Constraints To Participation In Outdoor Recreation Across The Life Span: A Nationwide Study Of Limitors And Prohibitors discusses the constraints of participating in outdoor recreational activities during the whole life cycle, which puts forward an opinion that there is a differentiation of the types of constraints during different period of the life cycle. Since there is also some research that mentions the leisure constraint theory in a long time period, this theory applies to long-term joint activities of the elderly and children in integrated welfare facilities. Also, the manuscript has briefly illustrated this part of the discussion above which is from line 151 to line 161:

Leisure constraint theory covers various types and a wide range of discussion. Most typical one in tourism studies is aimed to figure out the influencing factors of temporary activities during given organized tourists’ trip or group relaxing [19,20]. Besides, some scholars investigate factors long-termly affecting the participation of activities aimed at high school students in different grades [21], adults in different life cycles [22] and one group of people in different time period [23]. Since the joint activities in integrated welfare facilities exist in a long-term way, the decision making mode of this research is more close to the later research mode, so this research brings in the typical model of leisure constraint theory to discuss the decision making process of the joint activities of the elderly and children in integrated welfare facilities.

19. Tom, H,; Edgar, L.J.; Simon, H.; Gordon, W. Leisure Constraint Theory and Sport Tourism. Sport in Society. 2005, 8(2), 142-163. 10.1080/17430430500087435
20. Hinch, T.D.; Jackson, E.L.; Leisure Constraints Research: Its Value as a Framework for Understanding Tourism Seasonability. Current Issues in Tourism. 1995, 27(1), 87-106. 10.1080/00222216.1995.11969978
21. Jackson, E.L.; Rucks, V.C.; Negotiation of Leisure Constraints by Junior-High and High-School Students: An Exploratory Study. Journal of Leisure Research. 1995, 27(1), 85-105. 10.1080/00222216.1995.11969978
22. McGuire, F.A.; Dottavio, D.; O’Leary, J.T. Constraints to participation in outdoor recreation across the life span: A nation-wide study of limitors and prohibitors. The Gerontologist. 1986, 26(5), 538-544. https://doi.org/10.1093/geront/26.5.538
23. Jackson, E.L.; Witt, P.A. Change and Stability in Leisure Constraints: A Comparison of Two Surveys Conducted Four Years Apart. Journal of Leisure Research. 1994, 26(4), 322-336. 10.1080/00222216.1994.11969965

As for the conflicts caused by long-time interaction, this research has supplemental explanations for the particularity of integrated welfare facilities in “1. Introduction” from line 57 to 62:

By observation, these joint activities do not mean the elderly and children long-termly live together in one space, however, most of time, these two age groups of people are taken care of in their own functional spaces and carry on basic daily activities. Besides, planned by facility staff in a certain frequency, these joint activities are held randomly by borrowing some spaces in the elderly or children’s own functional spaces.

Since the interaction and touching between these two age groups are guided by volunteers and facilities staff in a short period of time instead of all day long, the conflicts caused by the long interaction you mentioned could be avoided.

- Joining the elderly and children is very dangerous from epidemic point of view. The social recursion of distributing the infection, for instance COVID-19, is “elderly-children-elderly”.

Response: It is of great importance to consider the influence of the infection especially COVID-19 you mentioned. In China, integrated welfare facilities are restrictedly controlled by the government which is represented not only in the aspect of the decision procedure of activities but also in the aspect of the prevention and treatment of infectious diseases. For example, the government also takes measures to prevent the spread of infectious diseases like influenza, by shutting down parks and quarantining. Take COVID-19 as an example, since the outbreak of COVID-19 in 2020, both of the facilities have been closed for a long time. In order to prevent infection in the park, people outside the facilities (such as volunteers, friends and relatives, etc.) are restricted to visit the parks when the epidemic is in serious situation. Therefore, the strict management of integrated welfare facilities has stopped the spread of the epidemic and further promoted the demand for social interaction in the park. So joint activities of the elderly and children have become more important to integrated welfare facilities. For the discussion above, the updated manuscript has added a corresponding explanation which is shown from line 615 to 621:

Especially in the situation of international pneumonia outbreak caused by the 2020 novel coronavirus disease (COVID-19), to avoid distributing the infection, China’s integrated welfare facilities all adopt close-end management measures. That means the social voluntary workers and even relatives are allowed to visit on a more limited frequency to different degrees. And this situation represents an even greater value of the joint activities in these facilities which have a promoting effect on the elderly and children’s social relationships and social interaction.

Besides, the current outbreak not only changed the status quo of joint activities, but also affect researchers’ methods. For example, at a moderate epidemic period, we can take the form of onsite observation and field interviews, but when the epidemic gets worse, we need to place cameras in facilities to collect relevant data after their permission, and further reduce the direct contact with children and the elderly. Therefore, in China, the epidemic has strengthened the control of integrated welfare facilities, bringing a more stable and safer environment for joint activities of the elderly and children in a closed environment.

The above has been revised responses to the manuscript. Your careful review has provided more help to our study clearer and more comprehensive. Thanks again for your advice!

Author Response

Dear reviewer:
Many thanks for your comments. The manuscript has been revised accordingly. The details are listed as below (please find the figure and table in the attachment):

1.    Main concerns

The article needs a short description of the limitations of the study. For instance: the low number of institutions considered, the small number of people who were surveyed, and so on.

Response: Under the premise that the previous research foundation is not sufficient, to some extent, it does lack sufficient research conditions and opportunities, leading to the limitations of the study. In Chapter 5 of the updated manuscript, we conclude two limitations. One is the problem of the small sample, and the other is the problem of a relatively simple selection of influencing factors of the model:

However, although the welfare development progress in China has been acknowledged throughout the world, as a developing country, it still needs further model clarification, mechanism development, space construction, and evaluation of management and operation. Therefore, it still lacks mature conditions and opportunities for discussion on the elderly and children’s joint activities in welfare facilities. And that is the main reason why this research prefers to be recognized as “pioneering research” and it shows research limitations in several aspects. For example, although there is a certain amount of staff working in integrated welfare facilities, only a few of them could participate in the decision procedure of joint activities planning. During the questionnaire survey, since it requires a certain amount of researchers to illustrate the definition and implication of influencing factors of decision making through semi-structured interviews, this research has small sample support which is a limitation to some extent. Although planning research has originally a small-sample characteristic [49], the data in this research represents a certain degree of value and meaning. But in future studies, relative researches could optimize the sample selection by choosing and screening more facility samples. (line 624 to 638)

Furthermore, there is a certain degree of “negotiation” in some specific activities which is also shown in the result of data analysis in this paper. Therefore, it is possible to further develop the research by optimizing the decision making model with a distillation of “negotiation factors” in future studies. (line 638 to 642)

49. Tang, W,F,; Big Data and Small Data: A Discussion on social science research methods. Journal of Sun Yat-sen University (Social Science Edition). 2015, 55(6), 141-146.DOI:10.13471/j.cnki.jsysusse.2015.06.014. (In Chinses)

The authors lack to mention the possible effects of the location´s architecture on the performance of the joint activities.

Response: This is a good suggestion. For the point you mentioned about “location´s architecture on the performance of the joint activities”, we regard it as the space attributes of jointing activities of the elderly and children such as accessibility, scale, openness, and so on. In fact, most of the authors of this article are in the architecture field, so the space attributes and design methods are our most concerned topics. However, in the process of proposing a set of spatial systems appropriate for the joint activities of the elderly and children, we found that there was a lack of relevant early basis, such as “what kind of activities should be held”, “how willing the elderly and orphans are to participate in the activities”, and “what kind of behavioral characteristics the participants have during the activities”. Therefore, we invited scholars from other fields to jointly complete this research on the decision making mechanism of joint activities for the elderly and children in integrated welfare facilities. The “location ´s architecture” you mentioned is the last part of our research on the topic of “joint activities for the elderly and children in integrated welfare facilities”. We considered this paper as the process of establishing an activity system and its decision making mechanism. The content about spatial characteristics you refer to is actually an important influencing factor in the construction of our “motivation-constraint interaction model” named “spatial support”, which refers to the spatial support conditions such as spatial scale, accessibility, openness and enclosure that meet the needs of joint activities (Table 2, line 293). During the semi-structured interview, we also explained the meanings of the “spatial support” indicator to the respondents in detail, which was basically consistent with what you pointed out. However, the motivation and constraint factors affecting joint activities of the elderly and children are too complex which are not on the same discussion level, we cannot discuss one of the influencing factors in detail at present. So we adopt qualitative evaluations. If the architectural conditions required by an activity are too strict, it will be eliminated due to the low value of evaluation in the aspect of spatial support during the activity screening procedure. In future research, we will further disassemble and deepen important indicators.

2.    Minor concerns

The authors used a model of leisure constrain theory for the decision making process of joint activities in China´s integrated facilities. However, they didn´t perform a direct survey to the decision makers or the elderly regarding the evaluation of the activities or about new options for the joint activities.

Response: Thanks for the meaningful question, and the issue you mentioned is just what we are working on at present. This manuscript focuses on the first part of a systematic study “the Construction Strategy of Joint Activity Space for the Elderly and Children in Integrated Welfare Facilities based on Decision Making and Activity Participation”. As for the research framework as well as the discussion of participants, the first and second authors of this manuscript have published an article entitled The Methods of Space Design in Integrated Care Welfare Facility Based on “Social Support” Theory on the top journal in the field of architecture in China named Architectural Journal in 2021. That article discussed the participants of the joint activities in detail especially the elderly who participate in the activities mainly for demands of “social support”. That is, they want to obtain physical and psychological satisfaction and comfort from the interaction with others. According to their demands, we put forward the “emotional-instrumental” social support framework to discuss their participation willingness in that article. 
The manuscript submitted this time focuses on the top-down “decision making” part of the joint activities, screening the activities according to feasibility levels, and analyzing the decision making mechanism of the joint activities. In the follow-up study, we will continue to explore the participation willingness and characteristics of participants with different social support demands in the scope of screened activities. The paper is limited by space so the research only discussed decision making mechanism of activities. Additionally, the willingness of the participants mentioned in your suggestion will be one complete research section in the future.

The abstract should include a brief description of the results.

Response: In response to your suggestion, we add a description of the results in the abstract (line 13 to 28) to make it clearer. The revised abstract is as follows:

In China, joint activities for the elderly and children in integrated welfare facilities lack systematic decision procedures. By learning from the “leisure constraint” theory, the study put forward 6 influencing indicators of motivation and constraint in the aspects of preliminary coordination, activity space and effect. By semi-structured interviews and questionnaire surveys analyzed by deviation value computation, the study analyzes the evaluation value of influencing factors in decision procedure of potential activity cases, where administrators and nurses act as two decision-makers. Further, it discusses the decision making mechanism based on “motivation-constraint” interaction model. Firstly, it analyses the dominated forces in decision procedure, which are “motivation oriented”, “negotiation oriented”, and “constraint oriented”. Secondly, it reveals that administrators and nurses as two decision-makers tend to give positive motivation evaluations and deliberative constraints evaluations respectively. And it analyzes the decision procedures of activities with distinct feasibility differentiation. Thirdly, it positions the levels of occurrence potential as “should occur”, “occurred but should be improved”, “potentially could occur”, and “hard to occur”. Eventually, it analyzes the requirements and potential for joint activities under different service modules, which provides a theoretical foundation for the systematic planning and development of the joint activities.

3.    Detailed review

Figure 1 needs a scale to appreciate the distances between welfare facilities.

Response: Thanks for your advice. We have added the complete map of China and added a scale on the map of Zhejiang Province to show the distances between welfare facilities in the maps in Figure 1 (line 51). 

The above has been revised responses to the manuscript. Your careful review has provided more help to our study clearer and more comprehensive. Thanks again for your advice!

Round 2

Reviewer 1 Report

Dear Authors,

This is a revised version of your original manuscript. You have corrected it according to the suggestions and it can be accepted without any changes now.

Best regards and good luck

Author Response

Dear Reviewer,

Your comments have greatly improved the structure and content of this manuscript. In the future, we will continue to focus on the joint activity study in the direction of participants' demands and the space design of buildings. Thanks again for your careful and kindly suggestions!

Best Wishes.